# Cyanidin-3-O-glucoside Contributes to Leaf Color Change by Regulating Two *bHLH* Transcription Factors in *Phoebe bournei*

**DOI:** 10.3390/ijms24043829

**Published:** 2023-02-14

**Authors:** Li Wang, Qiguang Wang, Ningning Fu, Minyan Song, Xiao Han, Qi Yang, Yuting Zhang, Zaikang Tong, Junhong Zhang

**Affiliations:** State Key Laboratory of Subtropical Silviculture, School of Forestry & Biotechnology, Zhejiang A&F University, Lin’an, Hangzhou 311300, China

**Keywords:** *P. bournei*, pigmentation, anthocyanin biosynthesis, transcription factor, transcriptome, metabonomic

## Abstract

Anthocyanins produce different-colored pigments in plant organs, which provide ornamental value. Thus, this study was conducted to understand the mechanism of anthocyanin synthesis in ornamental plants. *Phoebe bournei*, a Chinese specialty tree, has high ornamental and economic value due to its rich leaf color and diverse metabolic products. Here, the metabolic data and gene expression of red *P. bournei* leaves at the three developmental stages were evaluated to elucidate the color-production mechanism in the red-leaved *P. bournei* species. First, metabolomic analysis identified 34 anthocyanin metabolites showing high levels of cyanidin-3-O-glucoside (cya-3-O-glu) in the S1 stage, which may suggest that it is a characteristic metabolite associated with the red coloration of the leaves. Second, transcriptome analysis showed that 94 structural genes were involved in anthocyanin biosynthesis, especially flavanone *3′*-hydroxy-lase (*PbF3′H*), and were significantly correlated with the cya-3-O-glu level. Third, K-means clustering analysis and phylogenetic analyses identified *PbbHLH1* and *PbbHLH2*, which shared the same expression pattern as most structural genes, indicating that these two *PbbHLH* genes may be regulators of anthocyanin biosynthesis in *P. bournei*. Finally, overexpression of *PbbHLH1 and PbbHLH2* in *Nicotiana tabacum* leaves triggered anthocyanin accumulation. These findings provide a basis for cultivating *P. bournei* varieties that have high ornamental value.

## 1. Introduction

Leaf color is essential for the ornamental value of horticultural plants. Previous studies have shown that plant pigmentation is mainly attributed to the accumulation of anthocyanins, carotenoids, and chlorophyll, of which anthocyanins are the primary pigment responsible for the red color in leaves [1,2]. Cyanidin (Cya), pelargonidin (Pel), and delphinidin (Del) are the three most common anthocyanins found in nature and are responsible for the red, orange, and blue colors in plant tissues, respectively [3]. Plant anthocyanins are processed by glycosidic modification, and the most common glycoside modifications are glucose, galactose, rhamnose, xylose, arabinose, and fructose [4]. Glycosylation of anthocyanins improves their water solubility and stability and also serves as a signal for anthocyanins to undergo targeted transport to vesicles [5,6]. Glycosylation also has a significant effect on the color intensity produced by anthocyanins; for example, anthocyanins with a sugar molecule at the C3 site were observed to produce more intensely colored products than those with glycosylations at other positions [7]. Recently, an increasing number of anthocyanin modifiers have been discovered, and the main anthocyanin glycoside modifiers responsible for plant coloration are distinct in different plants. For example, the main anthocyanin modifiers responsible for fruit color were cya--3-O-rutinoside, cya-3-O-glucoside, and del-3-O-glucoside in *Kadsura coccinea* [8], while the main anthocyanins responsible for leaf color were malvidin-3-O-glucoside and petunidin-3-O-glucoside in *Solanum tuberosum* [9]. Therefore, elucidating the type and level of anthocyanin glycoside modifiers is a prerequisite for studying the coloring mechanism in plants.

Anthocyanins are synthesized through the flavonoid pathway under the regulation of several structural genes, which have been extensively studied in higher-order plants [10,11]. Anthocyanin biosynthesis begins with phenylalanine catabolism, which is catalyzed by phenylalanine ammonia-lyase (*PAL*) to cinnamic acid, followed by cinnamate 4 hydroxylase (*C4H*) and chalconeisomerase (*CHI*) to create 4-coumaroyl-CoA. Then, 4-coumaroyl-CoA is catalyzed by chalcone synthase (*CHS*) and chalcone isomerase (*CHI*) to naringenin, which is synthesized to Cya, Pel, and Del through flavanone *3*-hydroxy-lase (*F3H*), flavanone *3’*-hydroxy-lase (*F3’H*), flavanone *3’*, *5’*-hydroxy-lase (*F3’5’H*), dihydroflavonol 4-reductase (*DFR*), and anthocyanidin synthase (*ANS*) [12]. Finally, anthocyanins are glycosylated by UDP-glycose flavonoid glycosyltransferase (*UFGT*) and transferred to plant vesicles by glutathione S-transferase (*GST*) [13].

Previous studies have shown that these structural genes are synergistically regulated by the *MYB*-*bHLH*-*WD40* (MBW) complex [14]. For example, the significantly downregulated expression levels of *MYB* from the S1 to the S2 stage may limit the flavonoid-anthocyanin biosynthetic pathway, resulting in reduced anthocyanin accumulation in *Lagerstroemia indica* leaves [2]. Moreover, MrWD40-1 interacts with *MYB* and *bHLH* to enhance anthocyanin accumulation in *Myrica rubra* [15]. In addition, other transcription factors (TFs) such as *NAC* [16], *bZIP* [17], *WRKY*, and *ERF* [18] are involved in anthocyanin regulation. These results suggested that TFs and structural genes have essential functions in anthocyanin biosynthesis.

Recently, the combined analysis of the high-throughput transcriptome and metabolome has provided insight into anthocyanin transcriptional regulation and metabolic changes. For example, in *Cerasus humilis*, a combined metabolomic and transcriptomic analysis revealed that cyanidin-O-syringic acid and pelargonidin-3-O-beta-d-glucoside contribute to fruit peel color change by *ANS* and *UFGT* [19]. Cyanidin 3-O-rutinoside and Pelargonidin 3-O-rutinoside are the major anthocyanins that control fruit color, and *PtANS*, *PtUFGT*, and *PtGST11* are important structural genes for anthocyanin biosynthesis in *Prunus tomentosa* [20]. A similar study by Gao et al. showed that cyanidin 3,5-O-diglucoside is the main anthocyanin that contributes to red leaves and that *PAL*, *ANS*, *DFR*, and *F3H* are structural genes involved in leaf color production of *Acer pseudosieboldianum* [21]. There are also studies that show that the molecular mechanisms involved in leaf color changes vary in different species [22]. For example, two varieties of *Camellia sinensis* var. assamica, “Zijuan” and “Ziyan”, differ in the structural genes that regulate anthocyanin biosynthesis [23]. Therefore, it is necessary to study the molecular mechanisms behind leaf color change.

*P. bournei* is an evergreen tree of the Camphor family, an indigenous tree in China, which has high ornamental value due to its diversified variation in leaf color and upright trunk [24]. The evergreen tree canopy, rich leaf color, and excellent ornamental properties make it a famous ornamental tree species that has been increasingly applied to landscaping [25]. Among the germplasm resources of *P. bournei*, a large portion of the new leaves are red, but characteristics including the degree and duration of red color vary, and the mechanism underlying the production of red color is poorly understood. Recently, metabolomics has dramatically facilitated the identification of metabolic pathways of secondary metabolites in plants [26]. To understand the mechanism of color presentation of red-leaved *P. bournei*, metabolic changes were qualitatively and quantitatively analyzed during the color change of red-leaved *P. bournei* leaves using UPLC-MS/MS. The key anthocyanins that make its leaves appear red were explored, and the mechanism of leaf color change was elucidated. In the present study, we identified 57 anthocyanin components in *P. bournei* leaves at three developmental stages, of which cyanide-3-O-glucoside (cya-3-O-glu) might contribute to the red color in the new leaves. It also showed that *PbbHLH1* and *PbbHLH2* might participate in anthocyanin accumulation by upregulating anthocyanin biosynthesis-related genes. These findings provide a potential biotechnological strategy for selecting and breeding red-leaved *P. bournei* with high ornamental value.

## 2. Results

### 2.1. Phenotypic Characteristics of P. bournei Leaves

The new leaves of *P. bournei* trees are red, yellowish green, or light green and then gradually turn darker green as they develop (Figure 1A). Of them, red leaves are the most popular in China because of their bright color and their symbolic meaning of happiness. First, we analyzed the leaf color parameters (L*, a*, b*). The L* value represents the brightness of the leaf color. The a* value represents the red/green concentration of the color; a positive value of a* means red, and a negative a* value means green. The b* value indicates the concentration of yellow/blue elements. The analysis results show that there were significant differences between red and green leaves. Specifically, the L* value of red leaves was smaller than that of green leaves, showing that green leaves were brighter in color than red leaves. The a* value of red leaves was significantly higher than that of green leaves. The b* value increased significantly in the S3 stage of red leaves, while there was no change in green leaves (Figure 1B). Then, the levels of carotenoids, chlorophyll, and anthocyanins at three developmental stages were measured (Figure 1C). There were no significant differences in carotenoids among the three developmental stages. The total chlorophyll level of red-leaved samples was significantly lower than that of green-leaved samples, and the level increased from the S1 to S3 stages. The anthocyanin levels were significantly higher in red leaves than that in green leaves, and there was a significant decrease in anthocyanin levels as red leaves matured, suggesting that anthocyanin might be the main cause of the change in leaf color.

### 2.2. Anthocyanin Metabolites in the Red Leaves of P. bournei at Different Developmental Stages

To further elucidate the anthocyanin types and corresponding levels in the red leaves of *P. bournei* at three developmental stages, anthocyanin metabolites were analyzed by LC–MS/MS. The results identified 48 metabolites, including 15 cyanidin derivatives, 6 delphinidin derivatives, 2 malvidin derivatives, 3 petunidin derivatives, 3 pelnidin derivatives, 5 peonidin derivatives, 6 procyanidin derivatives, and 8 other flavonoids (Figure 2A, Appendix A). A hierarchical clustering analysis showed that the vast majority of metabolite levels were highest in the S1 stage and continually decreased in the S2 and S3 stages (Figure 2B). The metabolite profiles red *P. bournei* leaves were then subjected to principal component analysis (PCA). The PCA score plots exhibited an obvious separation among development stages. The first principal component (PC1) was 75.4%, and the anthocyanins present in the S1 stage were separated from those in the S2 and S3 stages (Figure 2C), indicating that the anthocyanin types and levels varied among the three developmental stages of the red *P. bournei* leaves.

### 2.3. Identification of Key Metabolites for Leaf Coloration

The orthonormal partial least squares regression–discriminant analysis (OPLS-DA) showed that the major metabolites varied significantly among developmental stages, with most metabolites highly accumulated in the S1 sample. It also revealed 26 anthocyanin glycosides were modified with variable importance in projection (VIP) values > 1 (Appendix A), all of which are characteristic substances of the S1 stage except for malvidin-3-O-glucoside (Figure 3A).

Among these eight anthocyanins, the level of cyanidin-3-O-glucoside (cya-3-O-glu) was much higher than that of the other anthocyanins (Figure 3B). High-performance liquid chromatography (HPLC) analysis showed that the peak levels of cya-3-O-glu in the leaves of the three developmental stages were consistent with the cya-3-O-glu standards and that the peak levels decreased with leaf development (Figure 3C). A standard curve was established using different standard concentrations of cya-3-O-glu to calculate the absolute amount of cya-3-O-glu in the samples (Figure 3D). The results showed that the levels of cya-3-O-glu decreased with leaf development, which was consistent with the metabolomic results (Figure 3E).

### 2.4. Transcriptomic Analysis of Leaves during Development

To further elucidate the molecular mechanism of anthocyanin biosynthesis in the red *P. bournei* leaves, leaves in three developmental stages (three biological replicates per stage) were sequenced, and nine cDNA libraries were constructed. After removing the adaptors and low-quality reads, more than 87% of the clean reads matched the *P. bournei* reference genome (NCBI under BioProject with accession number PRJNA627308) [27]. Overall, the RNA-seq data were of high quality and could be used for further analysis (Appendix A). A total of 28,218 unigenes were detected, resulting in 1528 upregulated and 1688 downregulated differentially expressed genes (DEGs) between the S1/S2 stages, 800 downregulated and 860 upregulated DEGs between the S2/S3 stages, and 2312 upregulated and 2349 downregulated DEGs between the S1/S3 stages (Figure 4A). A Venn diagram showed 548, 358, and 1447 unique DEGs between the S1/S2, S2/S3, and S1/S3 stages, respectively. There were 378 DEGs shared among the three groups, of which 191 DEGs showed a decreasing expression pattern from the S1 to S3 stage (Figure 4C).

Kyoto Encyclopedia of Genes and Genomes (KEGG) enrichment analysis showed that “Metabolic pathways (KO01100)” was the most enriched, followed by “Biosynthesis of secondary metabolites” (KO01110). Furthermore, anthocyanin biosynthesis (ko00942) and flavonoid metabolism (ko00941) were also significantly enriched (Figure 4B,D,E) (Appendix A).

### 2.5. DEGs Related to Anthocyanin Biosynthesis

Referencing the anthocyanin biosynthesis pathway in plants, we identified 11 structural genes (93 transcripts) involved in anthocyanin biosynthesis in *P. bournei* (Appendix A). In the early stages of anthocyanin biosynthesis, two *PAL* genes, one *C4H* gene, three *4CL* genes, three *CHS* genes, and one *CHI* gene were differentially expressed between the two stages. Specifically, the expression of two *PAL* genes (*PbPAL3* and *PbPAL4*), one *C4H* gene (*PbC4H2*), one 4*CL* gene (*Pb4CL7*), three *CHS* genes (*PbCHS1*, *PbCHS3*, and *PbCHS4*), and *PbCHI1* significantly decreased from the S1 to S3 stage (Figure 5).

The process of converting naringin to dihydrokaempferol is a key point in anthocyanin biosynthesis catalyzed by *F3H* genes. *PbF3H1* was highly expressed at the S1 stage, and the level gradually decreased with leaf development, while *PbF3H5* and *PbF3H10* were highly expressed at the S1 stage, with insignificant changes in the S2 and S3 stages, whereas the expression level of *PbF3H4* increased with leaf development.

*F3′H* and *F3′5′H* determine the anthocyanin precursor components. The expression of *F3′H* (*PbF3′H*) in S1 period was close to two times that in the S2 period and three times that in S3 period. The later stages of anthocyanin biosynthesis are also important for the accumulation of anthocyanins. *PbDFR* and *PbANS1* were highly expressed in the S1 stage and gradually decreased, showing significant differences in expression between the S2 and S3 stages. Two *UFGT* (*PbUFGT11* and *PbUFGT6*) genes were downregulated, and three *UFGT* (*PbUFGT1*, *PbUFGT2*, and *PbUFGT3*) genes were upregulated with leaf development. The expression levels of six *GTS* (*PbGST9*, *PbGST10*, *PbGST11*, *PbGST12*, *PbGST18*, and *PbGST27*) genes decreased significantly with leaf development. The level of *PbGST26* decreased significantly from the S1 stage to the S2 stage but increased significantly from the S2 stage to the S3 stage, while the level of *PbGST3* increased with leaf development.

### 2.6. Integrated Analysis of the Transcriptome and Metabolome

K-means analysis of all differentially expressed TFs and DEGs involved in the anthocyanin biosynthetic pathway was conducted, resulting in six K-means classes (Figure 6A). Among these classes, class 1 and class 2 contained 887 transcription factors with decreasing expression levels from the S1 stage to the S3 stage, which is consistent with the accumulation pattern of cya-3-O-glu and the S1-stage-signature differentially expressed metabolites (DAMs). These class 1 and class 2 transcription factors were mainly of the *MYB*, *bHLH*, and *ERF* gene families, and their expression patterns were consistent with those of structural genes (Appendix A). Of them, the *bHLH* family was the most predominant gene family, and a phylogenetic tree of the *bHLH* TF family was constructed using the *PbbHLH* members identified from the K-means classes 1/2 and known *bHLHs* involved in anthocyanin biogenesis in other plants (Appendix A). Combining the phylogenetic analyses and expression levels, two *bHLH* genes (*PbbHLH1* and *PbbHLH2*) were selected as candidate TFs that may be responsible for the regulation of anthocyanin biogenesis in *P. bournei* leaves.

Correlations between the transcriptional profiles of DEGs and S1-stage-characterized DAMs were analyzed using Pearson’s correlation coefficients (Appendix A). The result revealed that 30 DEGs involved in anthocyanin biosynthesis showed a significant correlation with at least two DAMs. Among the 30 DEGs, the expression levels of 10 DEGs (*PbF3H1*, *PbCHS3*, *PbCHS4*, *PbCHI1*, *Pb4CL7*, *PbF3′H*, *PbDFR*, *PbGST9*, *PbGST11*, and *PbUFGT6*) showed significant positive correlations with all of the DAMs, while five genes (*PbF3H4*, *Pb4CL13*, *PbUFGT1*, *PbUFGT2*, and *PbUFGT3*) showed negative correlations with only some of the DAMs (Figure 6B). Among these DEGs, *PbF3’H* specifically synthesizes dihydroquercetin, a precursor substance of cya-3-o-glu.

To further confirm the reliability of the RNA-Seq results, we performed Quantitative Real-time PCR (qRT–PCR) analysis of structural genes and predicted transcription factors involved in the anthocyanin biosynthetic pathway (Figure 6C,D). Both transcriptome and qRT–PCR analyses showed that the expression levels of *PbF3’H*, *PbbHLH1*, and *PbbHLH2* decreased with leaf development. The expression trend is the same between RNA-seq and qRT–PCR, indicating that the transcriptomic data were highly reliable and consistent with our prediction of transcription factors and structural genes.

Furthermore, the binding site of *bHLH* on the promoter region of *PbF3’H* was found (Figure 6E). Therefore, we suggest that in red-leaved *P. bournei* leaves, it is likely that the expression of *PbF3’H* is upregulated by *PbbHLH1* and *PbbHLH2*, resulting in the accumulation of cya-3-O-glu and leading to the reddening of leaves.

### 2.7. The Potential Regulatory Roles of PbbHLH1 and PbbHLH2

Using a well-established plant genetic transformation system, the *Agrobacterium tumefaciens* strain GV3101 containing 35S::*PbbHLH1* and 35S::*PbbHLH2* recombinant plasmids or an empty vector (EV) was injected into *N. tabacum* leaves, and multiple overexpression (OE) leaves were obtained. *Actin* was selected as the internal reference gene, and its relative expression was essentially the same in OE-*PbbHLH1* and OE-*PbbHLH2*, EV, and WT leaves (Figure 7A). The relative expression levels of the target genes *PbbHLH1* and *PbbHLH2* in the OE leaves were confirmed by semiqPCR (Figure 7B). To verify the regulatory roles of *PbbHLH1* and *PbbHLH2* in anthocyanin biosynthesis, the anthocyanin levels of *N. tabacum* leaves from OE, WT, and EV plants were measured. The OE leaves had significantly higher anthocyanin levels than the WT and EV leaves (Figure 7C).

The expression levels of nine structural genes (*NtF3′H*, *NtCHI*, *NtANS1*, *NtDFR1*, *Nt4CL1*, *NtPAL2*, *NtCHS5*, *NTGST*, and *NtUFGT*) were increased in the *PbbHLH1* OE leaves compared to the WT and EV leaves, and the expression levels of 10 structural genes (*NtF3′H*, *NtANS1*, *NtDFR1*, *Nt4CL1*, *NtPAL2*, *NtCHS3*, *NtCHS5*, *NtCHS6*, *NTGST*, and *NtUFGT*) were increased in the *PbbHLH2* OE leaves (Figure 7D). In summary, individually expressed *PbbHLH1* and *PbbHLH2* TFs are involved in anthocyanin accumulation by upregulating the expression levels of anthocyanin biosynthetic genes.

## 3. Discussion

### 3.1. Anthocyanin Characteristics Varied with Leaf Development in P. bournei

Anthocyanins, as critical secondary metabolites of plants, are synthesized through the flavonoid pathway and produce flowers, fruits, and leaves of plants with distinct colors such as pink, red, purple, and blue [10]. Recent studies have shown that anthocyanins are beneficial for the human body. For example, anthocyanin supplementation has a positive impact on human intestinal health [28], which also helps prevent cardiovascular disease and fight cancer [29]. In *Camellia japonica*, the rich anthocyanins not only make its petals greatly ornamental but also have high edible and medicinal value [30].

Previous studies have shown that color parameters give a suitable representation of the color differences among leaves [31]. For example, in the early stages of leaf development in *C. japonica*, the leaves appear red with a color parameter a* value of approximately 10, and the a* value decreases significantly when the leaves gradually turn green [32]. In this study, we found that the a* value was significantly higher in red leaves than that in green leaves. Consistently, the anthocyanin levels were significantly higher in red leaves than in green leaves, which decreased significantly with leaf development, indicating that anthocyanins might mainly contribute to the red color of the new leaves in *P. bournei*. A similar phenomenon was observed in *Boehmeria nivea*, in which the anthocyanin level in red-leaved variety was significantly higher than that of the green-leaved variety, indicating that anthocyanins are the main factor responsible for the variation in leaf color [31]. Several studies have shown that cyanidin, as one of six types of anthocyanins [33], contributes to red color in plants, but its composition and concentration vary significantly among plant species and/or tissues [34]. For example, in *Acer triflorum*, cya-3-O-arabinoside is closely associated with the red color in its leaves [22]. In *Saccharum officinarum*, cya-3-O-(6-O-malonyl)-glu plays a major role in the red coloration of its pericarp [35]. In the present study, 48 anthocyanins were identified in *P. bournei* leaves by metabolomics analyses, among which cya-based anthocyanins were predominant. However, unlike other plants, the anthocyanin with the highest level was cya-3-O-glu, especially in the S1 stage, as shown by the metabolomics and HPLC analysis. Although the S3 stage also contains common pelargonidin-, cyanidin-, and peonidin-based anthocyanin pigments, their levels may be too low to produce red pigmentation. Therefore, cya-3-O-glu might mainly contribute to red leaves in *P. bournei*.

### 3.2. The Genes Involved in Anthocyanin Biosynthesis in Red Leaves of P. bournei

In general, changes in secondary metabolite levels are consistent with changes in the abundance of structural gene transcripts in the corresponding biosynthetic pathways [36], and transcriptome sequencing provides the opportunity to analyze the expression of thousands of genes simultaneously [37]. In the present study, the combined transcriptome and metabolome analysis showed that the cascade pathway from phenylpropanoids to flavonoid compounds and, finally, anthocyanin biosynthesis was activated in the S1 stage and gradually decreased with leaf development, which may be a reasonable explanation for the higher anthocyanin level in the S1 stage. A total of 30 structural genes were identified as candidates for anthocyanin biosynthesis, of which 23 DEGs showed significantly higher expression levels in the S1 stage than in that in the S2 or S3 stage, showing a high correlation with the anthocyanin levels (Figure 7D). This result is consistent with several other studies that show that genes belonging to the early and late stages of anthocyanin synthesis regulate anthocyanin levels [38,39,40].

The phenylpropane pathway, which is the first stage of anthocyanin biosynthesis, is a common pathway for many of these metabolic processes. Previous studies have shown that anthocyanin accumulation is positively correlated with the structural genes *PAL*, *4CL*, and *C4H* in the phenylpropane pathway [41]. For example, low expression of *PAL* in white *Prunus persica* limits anthocyanin production [38]. In the present study, the expression of *PbPAL3*, *PbPAL4*, *PbC4H2*, and *Pb4CL7* was significantly higher in the S1 stage than in the S2 and S3 stages (*p* < 0.05), suggesting that the proteins they encode are likely to be the rate-limiting upstream enzymes of the anthocyanin metabolic pathway in *P. bournei*. Moreover, it was found that *PbPAL10* and *PbPAL13* may play different roles in the metabolic process. It has been reported that several *4CL* sequences may be specifically expressed and regulate metabolism only in certain tissues. For example, *Os4CL2* was specifically expressed in anthers and was strongly activated by UV irradiation in *Oryza sativa*, whereas other *4CL* members are not expressed in anthers [42]. Therefore, future experimental characterization will be performed to elucidate the differences among these *4CL* genes.

The second stage is the critical reaction stage of flavonoid metabolism. P-coumalic acid and 4-coumaroyl-CoA are catalyzed by *CHS* to synthesize chalcone, which is subsequently converted to naringenin by *CHI*, and then, naringenin is catalyzed by *F3H* to dihydrokaempferol [12]. A study found that the low expression levels of *CHS* and *F3H* in white petals reduced the dihydromyricetin (DHK) levels compared to those of red peach petals, thus inhibiting anthocyanin accumulation [43]. In this study, the expression levels of *PbCHS1*, *PbCHS3*, *PbCHS4*, *PbCHI1*, *PbF3H1*, *PbF3H5*, and *PbF3H10* in the S1 stage were significantly higher than those in the S2 and S3 stages (*p* < 0.01), which allowed more anthocyanin precursor dihydrokaempferol to proceed to the next synthesis step. The genes *F3’H*, *F3’5’H*, and *F3H* are key enzymes located at branching points that direct the pathway to produce red–purple, blue, and orange–red color, respectively [11]. It has been reported that the expression of *F3’H* in *Vitis vinifera* is involved in the regulation of anthocyanin biosynthesis [44]. In the present study, we suggested that the high expression of *F3’H* stimulates higher consumption of the precursor substance dihydrokaempferol for increased dihyquercetin synthesis and reduced synthesis of leucopelargonidin. This phenomenon may be the reason why more cyanidin-based anthocyanins are synthesized.

The third stage is the formation of various anthocyanins catalyzed by *DFR* and *ANS* [45]. Here, these anthocyanins encounter a series of glycosylation and methylation modifications, resulting in more stable anthocyanins being transferred to vesicles for storage via *GST* [46]. For example, strong anthocyanin accumulation in purple turnips was attributed to the upregulation of *DFR* genes [47]. Although *DFR* can substantially increase anthocyanin accumulation in plants, its substrate specificity causes the accumulation of different anthocyanins [48]. In our study, one differentially expressed *DRF* gene was identified in the leaves of red-leaved *P. bournei*. We speculate that the substrate preference of *PbDRF* may also be one of the reasons for the high cyanidin-based anthocyanin level, but this speculation remains to be verified by follow-up experiments.

In *Malus domestica*, the expression level of *ANS* determines the petal anthocyanin level [49]. A similar result was observed in the present study, in which high expression of *PbANS* was observed. In the Japanese apricot (*Prunus mume*), UFGT enzyme activity was found to increase when the petals appeared red, which is the same pattern as the accumulation of anthocyanins [50]. In the present study, we also found similar results to the above, where the high expression of *PbUFGT6* and *PbUFGT11* in the S1 stage promoted the synthesis of anthocyanins and increased their stability in plants. The reduction in anthocyanins in YW5AF7, a white-fruited variety of wild strawberry (*Fragaria vesca*), was attributed to a premature stop codon in the *glutathione S-transferase* (*GST*) gene [51]. In the present study, the high expression of *PbGST10*, *PbGST11*, *PbGST12*, *PbGST26*, and *PbGST27* in the S1 stage allowed more anthocyanins to be transferred to the vesicles, which led to the accumulation of more anthocyanins in red-leaved *P. bournei* in the S1 stage.

### 3.3. Transcription Factors Related to Anthocyanin Biosynthesis

Many studies have demonstrated that TFs play a key role in the anthocyanin biosynthetic pathway [19]. Of these TFs, *R2R3-MYB*, *bHLH*, and *WD40* individually or collectively contribute to the transcriptional regulation of the structural genes for anthocyanin biosynthesis [52]. For example, *MYB* and *bHLH* transcription factors control anthocyanins in *N. benthamiana* leaves [53]. In this study, we identified two *bHLH* genes, which were highly homologous to *bHLH* genes that regulate anthocyanin synthesis in other species that had significantly higher expression in the S1 stage than in the S2 and S3 stages of leaf development. By transiently overexpressing *PbbHLH1* and *PbbHLH2* in *N. tabacum* leaves, we found that *PbbHLH1* and *PbbHLH2* could upregulate the *PbF3′H* gene and other structural genes of the anthocyanin biosynthesis pathway, thus enhancing anthocyanin accumulation. A similar phenomenon was observed in the apple species *Malus domestica*, where overexpression of *MdbHLH51* in *Arabidopsis thaliana* increased anthocyanin content and upregulated the expression levels of structural genes related to anthocyanin biosynthesis [54]. The bHLH gene *NnTT8* was involved in the positive regulation of anthocyanin biosynthesis in *Nelumbo nucifera*, contributing to pigment production and providing a research basis for breeding of new plant varieties and nutraceutical development [55].

Gene transcriptional levels are influenced by cis-acting elements of their promoters, and different types and numbers of these elements lead to distinct gene expression patterns [56]. For many horticultural plants, ultraviolet radiation is necessary for anthocyanin biosynthesis in organs such as plant leaves [57]. In *P. bournei*, we found light-, temperature-, and hormone-related elements in the promoter regions of *PbbHLH1* and *PbbHLH2*. Therefore, it is speculated that these elements might lead to the elevated expression of the two *PbbHLHs* in the S1 stage, resulting in more anthocyanin accumulation. A similar result was observed in a pear species, in which the expression level of *PpbHLH64* was mediated by light at the transcriptional and posttranslational levels and positively regulated anthocyanin biosynthesis [58]. Temperature is also another major factor affecting anthocyanin biosynthesis, with low temperatures inducing anthocyanin synthesis and high temperatures inhibiting it for most plants [59]. For instance, the expression level of the bHLH gene *BrTT8* was significantly increased after low-temperature induction in the purple variety of *Brassica rapa*, resulting in increased anthocyanin biosynthesis [60].

Plant hormones are the main internal factors affecting anthocyanin accumulation [61]. Jasmonic acid (JA) is an important plant hormone, and in an apple species, the JAZ protein was found to interact with MYB and bHLH proteins to inhibit MBW protein complex formation, but this inhibition was reduced by JA treatment [62]. ABA is a key hormone for plants in response to various stresses and development and initiates the biosynthesis of plant anthocyanins in *Prunus* fruits [63]. In addition, auxins and cytokinins have also been found to influence anthocyanin biosynthesis. Moreover, the auxin signal elements AUX/IAAs and ARFs play key roles in regulating anthocyanin synthesis by directly interacting with the MBW complex and inhibiting its activity [64]. Therefore, we analyzed the cis-acting element of the *PbbHLH1* and *PbbHLH2* (Appendix A). We speculated that the two *PbbHLH* genes might play important roles in the anthocyanin biosynthesis of red-leaved *P. bournei*, and their expression levels might be regulated by light, temperature, and hormones. However, the regulatory mechanisms need further experimental verification.

## 4. Materials and Methods

### 4.1. Plant Materials

*P. bournei* plants were cultivated in the Qingyuan Experimental Forest Farm (118°24′ E, 29°08′ N), Lishui, China. Three 7-year-old “red” individuals of *P. bournei* were used to characterize the leaf color (Figure 1A). In April 2019, leaves from three developmental stages were used for transcriptome and metabolome analyses. Three biological replicates of each stage were performed, and samples were immediately frozen in liquid nitrogen and stored at ′80 °C.

### 4.2. Determination of Plant Pigments

#### 4.2.1. Determination of Total Anthocyanin Content (TAC)

TAC were extracted following the method described in [65] with some modifications. Specifically, 0.5 g of freeze-dried leaf powder was placed in 10 mL of methanol (containing 1% HCL) extract, shaken well, covered to protect it from light, and sonicated for 1 h. After centrifugation (5000 rpm, 10 min), the supernatant was used for anthocyanin content determination. Then, 2 mL of supernatant was aspirated and mixed with 8 mL of potassium chloride-hydrochloric acid buffer (pH 1.0) and sodium acetate-iceacetic acid buffer (pH 4.5), respectively, and the mixture was incubated in a dark room at 40 degrees for 30 min. The absorbance of mixture supernatant solutions was measured at 520 nm for anthocyanin content and 700 nm for haze correction using a UV–vis spectrophotometer (Thermo Scientific Co., Ltd., Wilmington, NC, USA) with 1 cm path length cuvettes. All absorbance measurements were performed at room temperature with distilled water as a blank control.

#### 4.2.2. Determination of Photosynthetic Pigments

The photosynthetic pigment content of the plants was extracted following the method described in [66] with some modifications. Specifically, 0.3 g of leaves was added to centrifuge test tubes, and 10 mL of ethanol-acetone (1:1, *v*:*v*) extraction solution was added so that the leaves were completely immersed in the liquid. The centrifuge tubes were capped and incubated in a dark oven at 30 °C until the leaves were completely decolored. The mixture was centrifuged (3000 rpm, 10 min), and the supernatant was used for the assay. The absorbance of the mixture supernatant was measured at 663, 645, and 470 nm using an UV–vis spectrophotometer (Thermo Scientific Co., Ltd., Wilmington, NC, USA) to determine the photosynthetic pigment content. All absorbance measurements were performed at room temperature using distilled water as a blank control.

### 4.3. Metabolomic Analysis of Anthocyanins

Metabolomic profiling was performed using the targeted metabolome method at Metware Biotechnology Co., Ltd. (Wuhan, Hubei, China). Briefly, leaves from three developmental stages were freeze-dried with a lyophilizer (100F, Scientz) and crushed with a mixer mill (MM400, Retsch) at 30 Hz for 1.5 min. Fifty milligrams of lyophilized leaf powder was dissolved in 500 μL of extraction solution (50% methanolic water containing 1% hydrochloric acid), vortexed for 10 min, sonicated for 10 min, and centrifuged (12,000 r/min) for 3 min [67]. The supernatant was aspirated and saved, and the process was repeated again. The two supernatants were combined and filtered through a 0.22 μm filter membrane, and then, the extracts were analyzed using Ultra Performance Liquid Chromatography-Tandem Mass Spectrometry (UPLC–MS/MS). A C18 column (1.7 µm, 2.1 mm*100 mm) was used for separation, and the sample injection volume was 2 μL. The conditions for metabolites were as follows: column temperature, 40 °C; flow rate, 0.2 mL/min. The gradient was composed of (A) ultrapure water containing 0.1% formic acid and (B) methanol containing 0.1% formic acid, and the linear gradient was as follows (mobile phase A:mobile B): 95:5 *v*/*v* for 0 min, 50:50 *v*/*v* for 6 min, 5:95 *v*/*v* for 12 min held for two minutes, and adjusted to 95:5 *v*/*v* equilibrium for two minutes at 14 min [68].

### 4.4. RNA Extraction, Library Construction, and Sequencing

Total RNA was extracted using Invitrogen’s PureLinkTM Plant RNA reagent (Thermo Scientific Co., Ltd., Wilmington, NC, USA), and RNA integrity was evaluated using RIN/RQN on an Agilent 2100. Then, the mRNA was enriched from total RNA by oligo (dT) magnetic beads, followed by the addition of fragmentation buffer to break the RNA into fragments, and first-strand cDNA was synthesized in the M-MuLv reverse transcriptase system. Subsequently, buffer, dNTP, and DNA polymerase I were added to synthesize the double-stranded cDNA. Then, 250–300 bp of adapter-linked cDNA fragments were purified using the AMPure XP system, and PCR was performed [69]. Finally, the PCR products were purified, and library quality was assessed using the Agilent Bioanalyzer 2100 system. The cDNA library was sequenced using the Illumina XtenPE150 system by Biomarker Technologies Co., Ltd. (Beijing, China).

### 4.5. RNA-Seq Analysis and Differentially Expressed Gene (DEG) Identification

Transcriptome sequencing was performed as described above. The reads from the Hiseq X Ten platform were mapped to the reference genomic sequences of *P. bournei* using HISAT2. The raw sequencing data of *P. bournei* were deposited in NCBI under BioProject accession number PRJNA627308 [27]. Then, gene annotation was performed using National Center for Biotechnology Information (NCBI) nonredundant protein sequences (NR), EuKaryotic Orthologous Groups (KOG), Kyoto Encyclopedia of Genes and Genomes (KEGG), and Gene Ontology (GO).

FPKM (Fragments Per Kilobase of transcript per Million fragments mapped) was used as a measure of transcript or gene expression level, and FPKM was calculated as follows: FPKM = mapped fragments of transcript/Total Count of mapped fragments* Length of transcript (kb). The identification of DEGs between two developmental stages was performed using the DESeq2 software package with a DFR (false-discovery rate) threshold of <0.05 and log2 (fold change) ≥ 1 [70,71]. The structural genes related to anthocyanin biosynthesis in *Arabidopsis thaliana* were used as the search sequences, and the candidate structural genes were obtained by a local blast search in the protein database of *P. bournei* (*p* < 1 × 10^−5^). The *Arabidopsis* reference genome was derived from https://www.arabidopsis.org/ (5 January 2021). The conserved structural domains were verified by referring to the NR, KOG, KEGG, and GO annotations and then using CD-search (https://www.ncbi.nlm.nih.gov/Structure/bwrpsb/bwrpsb.cgi (1 February 2021) in NCBI to finally obtain the structural genes of anthocyanin biosynthesis in *P. bournei*.

K-means clustering of structural genes and TFs for anthocyanin biosynthesis was performed using SPSS software and visualized using tools in the Metware cloud platform (cloud.metware.cn). Multiple sequence alignments of bHLH regulating anthocyanin biosynthesis in *Arabidopsis thaliana*, *Actinidia chinensis*, *Chrysanthemum × morifolium* and all bHLH in Class1 and 2 were performed using Clusta W.

A phylogenetic tree was constructed using maximum likelihood in MEGA X. Bootstraps were set to 1000, and the cqREV model was selected, thereby screening for differentially expressed TFs that might regulate anthocyanin biosynthesis in *P. bournei* [72].

### 4.6. Molecular Cloning of PbbHLH1 and PbbHLH2

RNA was extracted as described above, and cDNA was synthesized using HiScript^®^ III All-in-one RT SuperMix (Vazyme Co., Ltd., Nanjing, Jiangsu, China) and then used as a template for subsequent PCR amplification. Based on the *P. bournei* transcriptome data, two pairs of specific primers for PbbHLH1 and PbbHLH2 were designed. The gene-specific primers are listed in Appendix A. T-clone vectors with PbbHLH1 and PbbHLH2 sequences were used as templates for high-fidelity PCR amplification using 2x TransStart FastPfu PCR SuperMix (Tiangen Biochemical Technology Co., Ltd., Beijing, China). After the PCR products were detected by 1% agarose gel electrophoresis, the fragments were recovered and purified using a Midi Purification kit (Tiangen Biochemical Technology Co., Beijing, China). Using the ClonExpress^®^ II One Step Cloning Kit (Vazyme Co., Ltd., Nanjing, Jiangsu, China), PbbHLH1 and PbbHLH2 were cloned into the pK2GW7-eYGFPuv-3xFLAG overexpression vector with the Camv35S enhanced promoter and transformed into *Agrobacterium tumefaciens* strain GV3101.

### 4.7. Transient Expression Assay Using Nicotiana tabacum

Plates were cultured, and single colonies of *Agrobacterium tumefaciens* were selected for testing and incubated in YEP medium containing Rif and SPEC until they displayed an OD600 value of 0.6. One hundred milliliters of the resuspension solution containing acetosyringone, MgSO4, and MES was incubated until it displayed a OD600 of 0.6 and then was placed at room temperature for 1 h. Then, the *Agrobacterium tumefaciens* resuspension solution was inoculated on the surface of *N. tabacum* leaves with a syringe.

### 4.8. qPCR Analysis

Total RNA was extracted as above, and the purity and concentration were measured by a NanoDrop 2000 spectrophotometer (Thermo Scientific Co., Ltd., Wilmington, NC, USA). The first-strand cDNA was synthesized using the Prime-Script^®^ RT reagent kit (Takara, Dalian, Liaoning, China) according to the manufacturer’s guidelines. qPCR was performed using SYBR^®^ Premix Ex Taq II (Takara Co., Ltd., Dalian, Liaoning, China) on a CFX-96-well real-time PCR system (BioRad Co., Ltd., Hercules, CA, USA). The relative expression levels of target genes were calculated using the 2^-−ΔΔCT^ method and normalized using EF1α and actin reference genes in *P. bournei* and *N. tabacum*, respectively. The primers of target genes and reference genes are listed in Appendix A. Three replicates were performed for each sample.

### 4.9. Statistical Analysis

Log2 transformations were performed for the raw metabolomic data prior to further analysis. The data were analyzed by principal component analysis (PCA) and orthonormal partial least squares regression–discriminant analysis (OPLS-DA) using the SIMACA-P 13.0 software package [73]. Heatmap performed using the Metware Cloud, a free online platform for data analysis (https://cloud.metware.cn (accessed on 10 August 2022)). All data were analyzed using SPSS and displayed as the mean ± standard deviation (SD) (*n* = 3). The data were analyzed using one-way ANOVA and Duncan’s test, with significance set at *p* < 0.05. Pearson correlation analysis was performed, and these relationships were visualized using Cytoscape software (version 3.8.2) [74].

## 5. Conclusions

Collectively, the mechanisms underlying anthocyanin accumulation in *P. bournei* leaves were analyzed by combined metabolomic and transcriptomic analyses. At least 26 anthocyanins exhibited significant differences among leaves in different developmental stages, and cya-3-O-glu was identified as the major anthocyanin present in red leaves. Moreover, anthocyanin biosynthetic genes were identified, and co-expression analysis suggested that PbF3’H is the key gene involved in cya-3-O-glu accumulation. Moreover, two bHLH genes (*PbbHLH1* and *PbbHLH2*) are involved in the anthocyanin synthesis pathway and may regulate the expression of structural genes for anthocyanin biosynthesis in *P. bournei*. Furthermore, the overexpression of *PbbHLH1* and *PbbHLH2* in *N. tabacum* leaves increased the anthocyanin levels and upregulated the expression levels of structural genes. This study improves our understanding of anthocyanin accumulation and the molecular mechanism underlying anthocyanin biosynthesis in *P. bournei* leaves. However, whether the transcription factors PbbHLH1 and PbbHLH2 work together with other transcription factors to regulate other structural groups of anthocyanin biosynthesis, thereby promoting anthocyanin accumulation, needs further investigation.

## Figures and Tables

**Figure 1 ijms-24-03829-f001:**
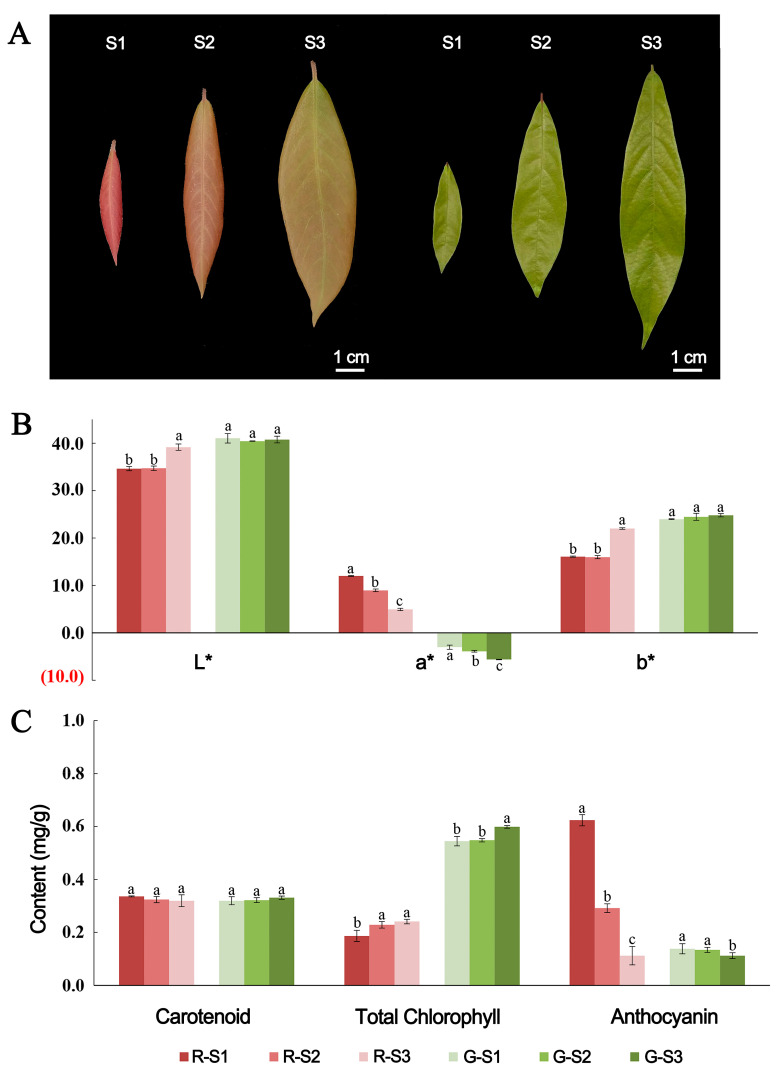
Leaf color parameters and pigment level of red-leaved *P. bournei* and green-leaved *P. bournei.* (**A**) Leaves of three developmental stages of different color *P. bournei*; (**B**) Analysis of Variance (ANOVA) on leaf color parameters. Means of different treatment groups (different lowercase letters indicate significant differences) had a significant difference at *p* ≤ 0.05. (**C**) Chromophore (carotenoids, chlorophylls, anthocyanins) levels and contrast analysis. Means of different treatment groups (different lowercase letters indicate significant differences) had a significant difference at *p* ≤ 0.05.

**Figure 2 ijms-24-03829-f002:**
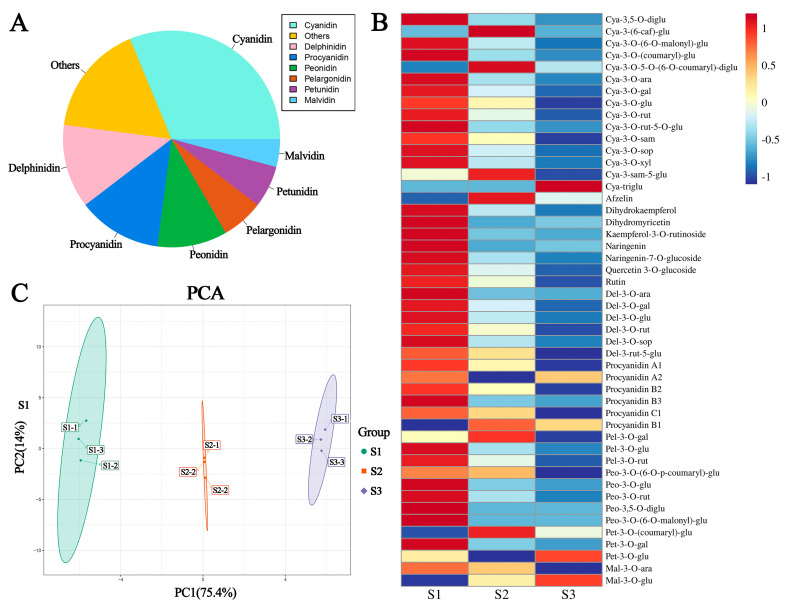
Analysis of anthocyanin levels in three developmental stages of red *P. bournei* leaves. (**A**) Metabolites detected in leaves at three developmental stages; (**B**) heatmap analysis of all detected metabolites; (**C**) the principal component analysis (PCA) plot illustrates clear discrimination among metabolites in the three developmental stages.

**Figure 3 ijms-24-03829-f003:**
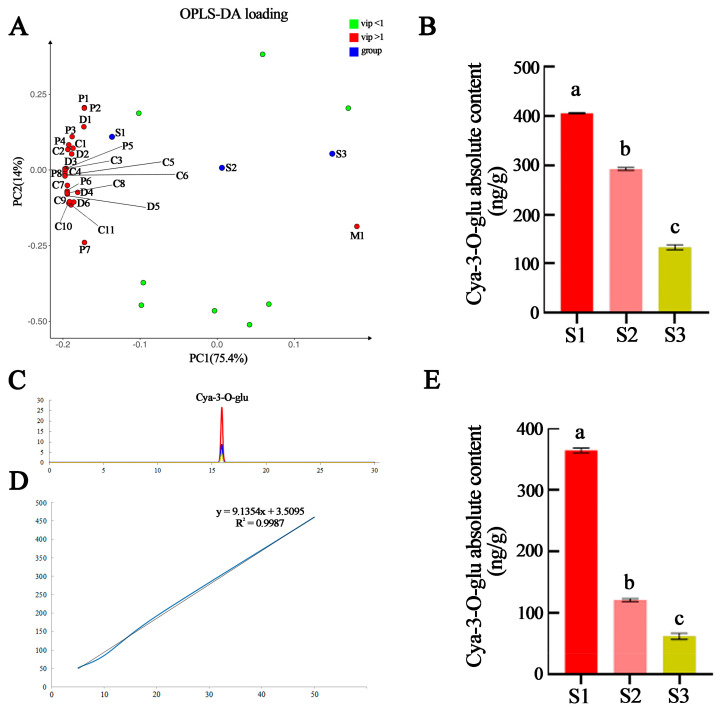
Identification of key metabolites between red *P. bournei* leaves in three developmental stages. (**A**) The The orthonormal partial least squares regression–discriminant analysis (OPLS-DA) loading plot of anthocyanin levels of the leaves in three developmental stages. P1, peonidin 3,5-O-diglucoside (peo-3,5-diglu); P2, peonidin-3-O-(6-O-p-coumaryl)-glucoside (peo-3-O-(6-O-malonyl)-glu); P3, petunidin 3-O-arabinoside (pet-3-O-ara); P4, peonidin 3-O-glucoside (peo-3-O-glu); P5, peonidin 3-O-rutinoside (peo-3-O-rut); P6, pelargonidin 3-O-rutinoside (pel-3-O-rut); P7, peonidin-3-O-(6-O-p-coumaryl)-glucoside (peo-3-O-(6-O-coumaryl)-glu); P8, pelargonidin 3-O-glucoside (pel-3-O-gly); D1, delphinidin 3-O-arabinoside (del-3-O-ara); D2, delphinidin 3-O-sophoroside(del-3-O-sop); D3, delphinidin 3-O-galactoside (del-3-O-gla); D4, delphinidin 3-O-glucoside (del-3-O-glu); D5, delphinidin 3-O-rutinoside (del-3-O-rut); D6, delphinidin-3-rutinoside 5-glucoside (del-3-rut-5-glu); C1, cyanidin-3-O-(coumaryl)-glucoside (cya-3-O-(coumaryl)-glu); C2, cyanidin-3-rutinoside 5-glucoside (cya-3-O-rut-5-O-glu); C3, cyanidin 3-O-arabinoside (cya-3-O-ara); C4, cyanidin 3,5-O-diglucoside (cya-3,5-O-diglu); C5, cyanidin-3-O-xyloside (cya-3-O-xyl); C6, cyanidin 3-O-(6-O-malonyl-beta-D-glucoside) (cya-3-O-(6-O-malonyl)-glu); C7, cyanidin 3-O-galactoside (cya-3-O-gal); C8, cyanidin 3-O-rutinoside (cya-3-O-rut); C9, cyanidin 3-O-glucoside (cya-3-O-glu); C10, cyanidin 3-O-sophoroside (cya-3-O-sop); C11, cyanidin 3-O-sambubioside (cya-3-O-sam); M1, malvidin 3-O-glucoside (mal-3-O-glu). (**B**) Absolute levels of cya-3-O-glu were measured by anthocyanin-targeted metabolism analysis. Means of different treatment groups (different lowercase letters indicate significant differences) had a significant difference at *p* ≤ 0.05; (**C**) chromatograms of cya-3-O-glu in *P. bournei* leaves at different developmental stages as determined by High-performance liquid chromatography (HPLC); (**D**) standard curve created using cya-3-O-glu standard concentrations; (**E**) absolute levels of cya-3-O-glu measured by HPLC. Means of different treatment groups (different lowercase letters indicate significant differences) had a significant difference at *p* ≤ 0.05.

**Figure 4 ijms-24-03829-f004:**
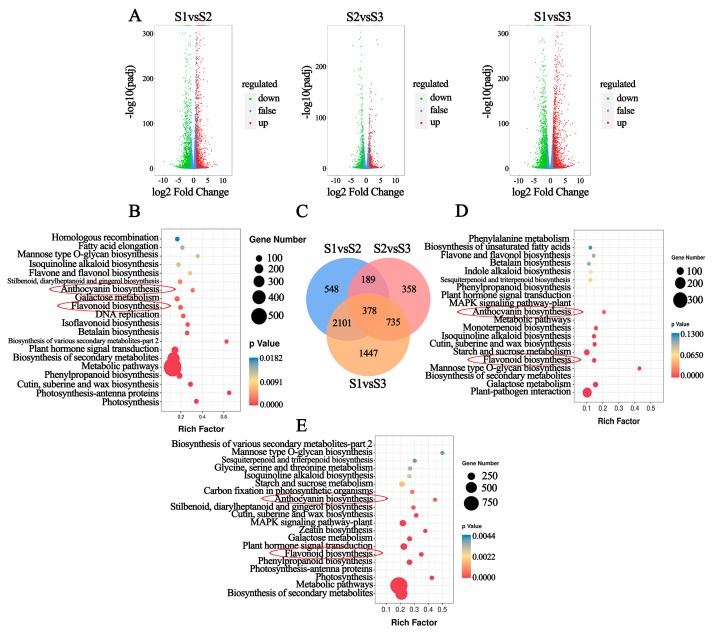
Identification and Kyoto Encyclopedia of Genes and Genomes (KEGG) enrichment of differentially expressed genes (DEGs) in three developmental stages of red *P. bournei* leaves. (**A**) Volcano plots of DEGs; green, red, and blue dots represent the downregulated, upregulated, and non-differentially expressed genes, respectively; (**C**) Venn diagram of DEGs in each comparison group. KEGG enrichment analysis of the DEGs between; (**B**) S1 vs. S2; (**D**) S2 vs. S3 and (**E**) S1 vs. S3. The red circle highlights the flavonoid biosynthetic process and anthocyanin biosynthetic process.

**Figure 5 ijms-24-03829-f005:**
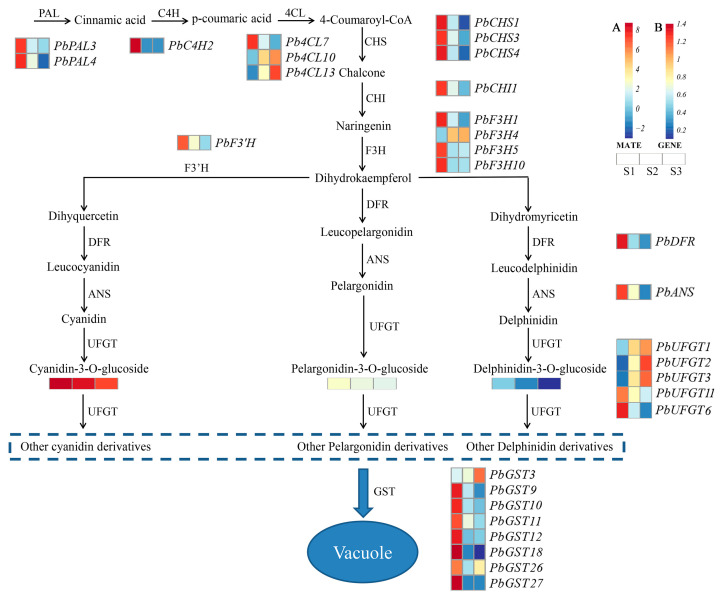
Reconstruction of anthocyanin biosynthetic pathways during leaf development using differentially expressed structural genes. (**A**) Anthocyanin levels at three developmental stages; (**B**) expression of differentially expressed structural genes at three developmental stages. *PAL*, phenylalanine ammonia-lyase; *C4H*, Cinnamate 4-hydroxylase; *4CL*, 4-coumarate-CoA ligase; *CHS*, chalcone synthase; *CHI*, chalcone isomerase; *F3H*, flavanone 3-hydroxylase; *F3′H*, Flavonoid 3′-hydroxylase; *DFR*, dihydro flavonol reductase; *ANS*, anthocyanidin synthase; *UFGT*, UDP-glycose flavonoid glycosyltransferase; *GST*, glutathione s-transferase.

**Figure 6 ijms-24-03829-f006:**
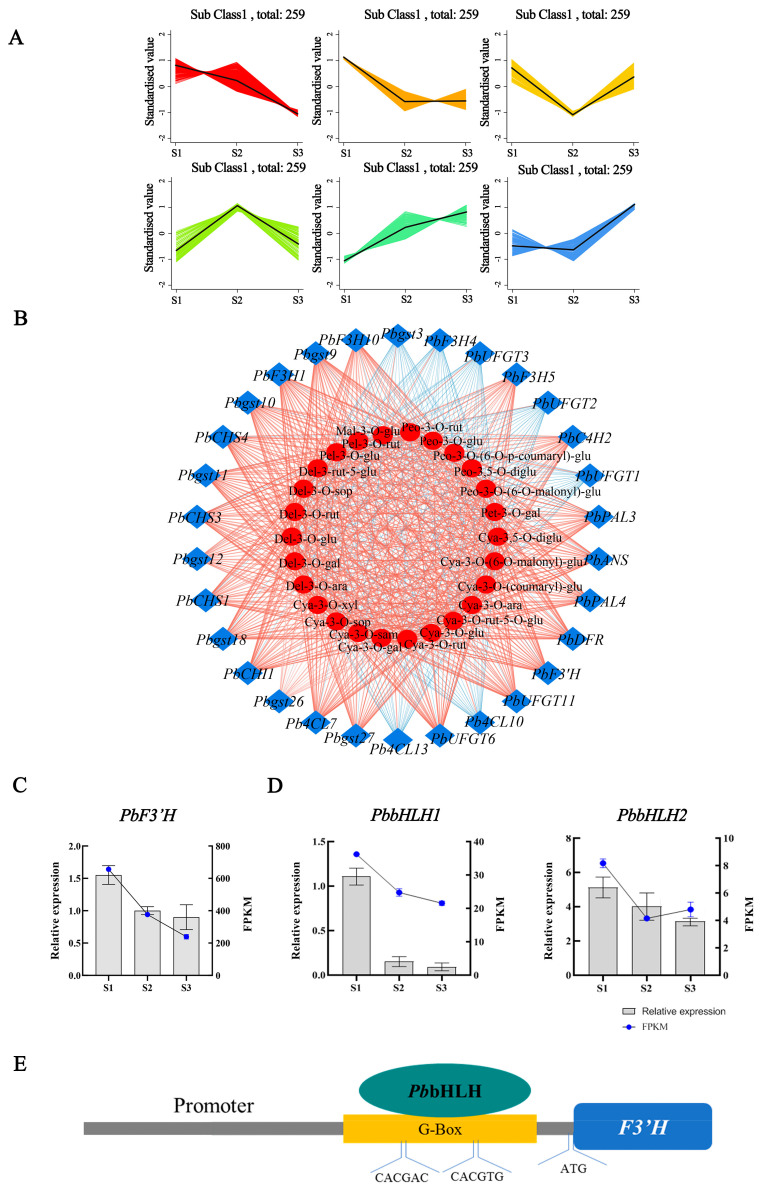
Comprehensive analysis of the transcriptome and metabolome and identification of key structural genes and transcription factors. (**A**) Six subclasses of co-expressed genes and their kinetic patterns during leaf development. (**B**) Correlation analysis of DEGs and differentially expressed metabolites (DAMs). The red line indicates a positive correlation, and the blue line indicates a negative correlation. The line thickness represents the correlation index, and correlations with R > 0.8 and *p* < 0.05 were deemed significant. (**C**,**D**) The expression patterns of *PbF3′H*, *PbbHLH1*, and *PbbHLH2* between leaves in three developmental stages were evaluated. Quantitative Real-time PCR (expression levels were calculated for *PbF3′H*, *PbbHLH1*, and *PbbHLH2* and *EF1α* expression was used for normalization in *P. bournei*. Each value is the mean of three replicates, and the error line indicates the standard deviation. (**E**) Analysis of *cis-acting* elements of the promoter of *PbF3′H* in *P. bournei*.

**Figure 7 ijms-24-03829-f007:**
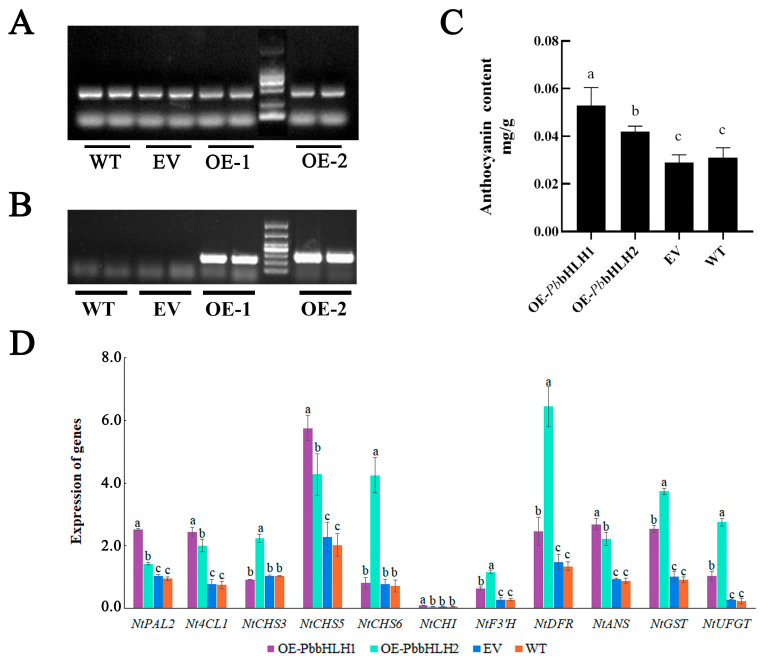
Overexpression of *PbbHLH1* and *PbbHLH2* increased anthocyanin accumulation in *N. tabacum* leaves. (**A**) Consistent expression of the internal reference gene *actin* in the WT, EV, and OE leaves as determined by semiqPCR. (**B**) Relative expression levels of *PbbHLH1* and *PbbHLH2* in the WT, EV, and OE leaves by semiqPCR. (**C**) The anthocyanin levels in the *PbbHLH1* and *PbbHLH2*-overexpressing leaves of *N. tabacum*. At least three independent experiments were performed for each sample, and data are shown as the mean ± SD. Means of different treatment groups (different lowercase letters indicate significant differences) had a significant difference at *p* ≤ 0.05; (**D**) The expression levels of anthocyanin biosynthesis-related genes in the WT, EV, OE1, and OE2 leaves of *N. tabacum* leaves. The significant difference threshold was set at *p* < 0.05. Means of different treatment groups (different lowercase letters indicate significant differences) had a significant difference at *p* ≤ 0.05.

## Data Availability

Not applicable.

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
