# Peer review of "Cyanidin-3-O-glucoside Contributes to Leaf Color Change by Regulating Two bHLH Transcription Factors in Phoebe bournei"

_ijms, 2023, doi:10.3390/ijms24043829_

Round 1

Reviewer 1 Report

The manuscript entitled “Cyanidin-3-O-glucoside contributes to leaf color change by 2 regulating two bHLH transcription factors in Phoebe bournei” is very interesting and well written. I have some comments and suggestions for further improvements.

1.     Abstract: line 12; consider replacing “Thus, we are interested in understanding the mechanism of anthocyanin synthesis in ornamental plants” sentence with Thus, this study was conducted to understand the mechanism of anthocyanin synthesis in ornamental plants.

2.     Line 36 use pigmentation instead of coloration.

3.     Line 450: (Figure S3) please check throughout the manuscript.

4.     Line 569: check if this is the correct way to write it 2 −ΔΔCt

5.     However, the precise mechanism requires further studies? Specify your suggestion and what do you mean by precise mechanism?

Author Response

Response to Reviewer 1 Comments

Thank you for handling the review process and for the reviewers’ comments concerning our manuscript entitled “Cyanidin-3-O-glucoside contributes to leaf color change by regulating two bHLH transcription factors in Phoebe bournei”. These comments are valuable and very helpful. We have studied the comments carefully and have made corrections that we hope meet with approval. Based on the instructions provided in your letter, we uploaded the file of the revised manuscript. We have made an extensive revision to improve the presentation at all levels of the manuscript and have made the description of the methods more accurate and detailed. We have tried our best to carefully respond to the reviewer’s comments. 

Point 1: Abstract: line 12; consider replacing “Thus, we are interested in understanding the mechanism of anthocyanin synthesis in ornamental plants” sentence with Thus, this study was conducted to understand the mechanism of anthocyanin synthesis in ornamental plants.

Response 1: DONE. Thank you for your suggestion. We have revised to “Thus, this study was conducted to understand the mechanism of anthocyanin synthesis in ornamental plants”.

Point 2: Line 36 use pigmentation instead of coloration.

Response 2: DONE. Thank you for your suggestion. We have revised to “pigmentation”.

Point 3: Line 36 use pigmentation instead of coloration.

Response 3: DONE. Thank you for your suggestion. We have checked the Figure S3 was in line 481.

Point 4: Line 569: check if this is the correct way to write it 2 −ΔΔCt

Response 4: DONE. We are sorry for our carelessness, and we have revised to 2 -ΔΔCT.

Point 5: However, the precise mechanism requires further studies? Specify your suggestion and what do you mean by precise mechanism?

Response 5: DONE. Thank you for your suggestion. We have revised to “whether the transcription factors PbbHLH1 and PbbHLH2 work together with other transcription factors to regulate other structural genes, thereby promoting anthocyanin accumulation, needs further investigation.

Reviewer 2 Report

In this study, authors conducted metabolic and transcriptomic analyses of Phoebe bournei red leaves at three different stages to reveal color production mechanisms in the red leaves of P. bournei. Authors identified high level of cyanidin-3-O-glucoside at the S1 stage and it is also highly correlated with anthocyanin biosynthesis. Transcriptome data, clustering analyses, and overexpression experiment revealed that two bHLH transcription factor play an important role for anthocyanin accumulation. Overall, authors provide enough results for their research purposes. Manuscript is fine publication after minor revision. I have several comments as follows.

Abstract is relatively long. The maximum length of abstract should be less than 200 words. Please revise the abstract.

L15-17 Authors presented metabolic data firstly. Therefore, please correct the sentence.

L16 in the three -> at the three

L76-78 It is recommended to indicate the name of plant for the reference 22.

L85 Indicate the name of plant species which authors referred.

Introduction was nicely written. However, I feel that the introduction is relatively long. Please some unnecessary paragraphs.

L112-122 Please explain three leaf color parameters in the beginning of the paragraph. Please reorganize this paragraph.

The image quality for Figure 1 should be improved. Please use “Arial” font for figures.

L132-136 Please describe about a, b derived from ANOVA test.

Figure 2 should be replaced with a high-quality image. Please increase size of

Figure 2A and 2C.

L152-155 Please indicate programs to generate heatmap and PCA plot.

L157 Please define OPLS-DA loading analysis. Nobody knows.

L159 Spell out “VIP” and define it.

Figure 3 is also too small and timid. Please replace images in the Figure 3.

L182 Please indicate the website for P. bournei reference genome.

Images in the Figure 4 should be magnified and improved. Too small to read. The order of images should be checked. Not A, C, B, D, E

For subsection 2.4, I suggest presenting only following comparison.

S2 vs. S1 and S3 vs. S1

Authors should indicate which stage was used as a control for comparison to get DEGs. Too many comparisons make results confusing to understand.   

Figure 5 is fine. Just increase size of images.

Figure 6 and 7 are again not readable. Please replace the images with high quality.

In discussion, all subtitle should be removed.

Materials and methods

Please describe detailly for materials. For example (Thermo Scientific Co., Ltd., City, USA) Name of city should be included. Check for other materials.

L505-506 Authors used Invitrogen’s product and referred Thermo Fisher. Not logical

RNA-seq data analyses were not written in detail. Please provide programs and methods for mapping, reference genome, calculation of expression, identification of DEGs. Sources of various databases such as NR, KOG, KEGG, and GO should be described.

L554 Transient transformation ~~ -> Transient expression assay using Nicotiana tabacum

Author Response

Response to Reviewer 2 Comments

Thank you for handling the review process and for the reviewers’ comments concerning our manuscript entitled “Cyanidin-3-O-glucoside contributes to leaf color change by regulating two bHLH transcription factors in Phoebe bournei”. These comments are valuable and very helpful. We have studied the comments carefully and have made corrections that we hope meet with approval. Based on the instructions provided in your letter, we uploaded the file of the revised manuscript. We have made an extensive revision to improve the presentation at all levels of the manuscript and have made the description of the methods more accurate and detailed. We have tried our best to carefully respond to the reviewer’s comments. 

Point 1:  Abstract is relatively long. The maximum length of abstract should be less than 200 words. Please revise the abstract.

Response 1: DONE. Thank you for your suggestion. We have reduced the Abstract to 199 words.

Point 2: L15-17 Authors presented metabolic data firstly. Therefore, please correct the sentence.

Response 2: DONE. Thank you for your suggestion. We have revised to “the metabolic data and gene expression of red P. bournei leaves at the three developmental stages were evaluated to elucidate the color production mechanism in the red-leaved P. bournei species.

Point 3: L16 in the three -> at the three

Response 3: DONE. Thank you for your suggestion. We have revised to “at the three”.

Point 4: L76-78 It is recommended to indicate the name of plant for the reference 22.

Response 4: DONE. Thank you for your suggestion. We have indicated the name of the plant for the reference 22 in line 81-84.

Point 5: L85 Indicate the name of plant species which authors referred.

Response 5: DONE. Thank you for your suggestion. We have revised to “A similar study by Gao et al., showed that Cya-3,5-O-dig is the main anthocyanin that contributes to red leaves, and that PAL, ANS, DFR, and F3H are structural genes involved in leaf color production of Acer pseudosieboldianum”.

Point 6: Introduction was nicely written. However, I feel that the introduction is relatively long. Please some unnecessary paragraphs.

Response 6: DONE. Thank you for your suggestion. We have revised the Introduction by removing some unnecessary paragraphs.

Point 7: L112-122 Please explain three leaf color parameters in the beginning of the paragraph. Please reorganize this paragraph.

Response 7: DONE. Thank you for your suggestion. This section has been revised according to the information showed in the work suggested by the reviewer (Line 128-140).

Point 8: The image quality for Figure 1 should be improved. Please use “Arial” font for figures.

Response 8: DONE. Thank you for your suggestion. We have improved the quality of Figure 1 and replaced the font with "Arial".

Point 9: L132-136 Please describe about a, b derived from ANOVA test.

Response 9: DONE. Thank you for your suggestion. We have described “a, b” derived from ANOVA test in line 152-155.

Point 10: Figure 2 should be replaced with a high-quality image. Please increase size of Figure 2A and 2C.

Response 10: DONE. Thank you for your suggestion. We have improved the quality of Figure 2 and increased the size of Figure 2A and 2C. Now the resolution of Figure 2 is 500 dpi.

Point 11: L152-155 Please indicate programs to generate heatmap and PCA plot.

Response 11: DONE. Thank you for your suggestion. We have added the description on generating heatmap and PCA plot in Materials and Methods, in L160-621.

Point 12: L157 Please define OPLS-DA loading analysis. Nobody knows.

Response 12: DONE. We deeply appreciate the reviewer’s suggestion. According to the reviewer’s comment, we have added a more detailed interpretation regarding OPLS-DA analysis in line L177-178.

Point 13: Spell out “VIP” and define it.

Response 13: DONE. Thank you for your suggestion. We have defined the “VIP” in the L180-181.

Point 14: Figure 3 is also too small and timid. Please replace images in the Figure 3.

Response 14: DONE. Thank you for your suggestion. We have improved the quality of Figure 3 and made adjustments to its size.

Point 15:  Please indicate the website for P. bournei reference genome.

Response 15: DONE. Thank you for your suggestion. We have added the reference genome in L210 and L554-555.

Point 16: Images in the Figure 4 should be magnified and improved. Too small to read. The order of images should be checked. Not A, C, B, D, E

Response 16: DONE. Thank you for your suggestion. We have improved the quality of Figure 4, resized the images, and renumbered them. Now the resolution of Figure 4 is 500 dpi.

Point 17: For subsection 2.4, I suggest presenting only following comparison. S2 vs. S1 and S3 vs. S1. Authors should indicate which stage was used as a control for comparison to get DEGs. Too many comparisons make results confusing to understand.

Response 17: Thank you for your suggestion. If we presented only two comparisons, such as S2 vs. S1 and S3 vs. S1, the results seemed more clearly. However, many important information, such as these DECs involving in S2 vs. S3, will be lost. Therefore, we hope to keep all three comparisons.

Point 18: Figure 5 is fine. Just increase size of images.

Response 18: DONE. Thank you for your suggestion. We have increased the size of images in Figure 5.

Point 19: Figure 6 and 7 are again not readable. Please replace the images with high quality.

Response 19: DONE. Thank you for your suggestion. We have increased the size of Figure 6 and 7, with a resolution of 500dpi.

Point 20: In discussion, all subtitle should be removed.

Response 20: Thank you for your suggestion. However, in order to presenting three main points more clearly in the Discussion part, the subtitles should be kept. We also found that most publications in IJMS have the subtitles in the Discussion.   

Point 21: Please describe detailly for materials. For example (Thermo Scientific Co., Ltd., City, USA) Name of city should be included. Check for other materials.

Response 21: DONE. Thank you for your suggestion. We have improved these descriptions.

Point 22: L505-506 Authors used Invitrogen’s product and referred Thermo Fisher. Not logical

Response 22: DONE. Thank you for your suggestion. The takeover of Invitrogen Ltd. has been done by Thermo Ltd. and so we referred Thermo Ltd.

Point 23: RNA-seq data analyses were not written in detail. Please provide programs and methods for mapping, reference genome, calculation of expression, identification of DEGs. Sources of various databases such as NR, KOG, KEGG, and GO should be described.

Response 23: DONE. Thank you for your suggestion. We have improved the description on the RNA-seq data analysis.

Point 24: L554 Transient transformation ~~ -> Transient expression assay using Nicotiana tabacum

Response 24: DONE. Thank you for your suggestion. We have revised to “Transient expression assay using Nicotiana tabacum” in L596.
